# Shrub Cover and Soil Moisture Affect *Taxus baccata* L. Regeneration at Its Southern Range

**DOI:** 10.3390/plants12091819

**Published:** 2023-04-28

**Authors:** Giacomo Calvia, Paolo Casula, Emmanuele Farris, Giuseppe Fenu, Sergio Fantini, Gianluigi Bacchetta

**Affiliations:** 1Centre for the Conservation of Biodiversity (CCB), Department of Life and Environmental Sciences, University of Cagliari, Viale Sant’Ignazio da Laconi 13, 09123 Cagliari, Italy; gfenu@unica.it (G.F.); sfantini@forestas.it (S.F.); bacchet@unica.it (G.B.); 2Servizio Tecnico, Agenzia Forestas, Viale Luigi Merello 86, 09123 Cagliari, Italy; pcasula@forestas.it; 3Department of Chemical, Physical, Mathematical and Natural Sciences, University of Sassari, Via Piandanna 4, 07100 Sassari, Italy; emfa@uniss.it; 4NBFC, National Biodiversity Future Center, 90133 Palermo, Italy

**Keywords:** common yew, EU habitat, Mediterranean, overbrowsing, recruitment, shrubs, water availability

## Abstract

The effect of key ecological and anthropic factors on the recruitment of the common yew (*Taxus baccata* L.) in Sardinia (Italy) has been analyzed. After bibliographic and cartographic research, followed by field surveys, we found 232 sites where yew grows in Sardinia (as opposed to 69 previously reported in the literature). Among them, we selected 40 sites, located in 14 different mountain chains, characterized by a number of individuals ranging from 11 to 836 adult yews with an average diameter at breast height (DBH) from 13 to 130 cm. By means of generalized linear modeling, we investigated and weighted the effect of ecological, structural, and anthropic factors on the amount of *T. baccata* recruitment. Our analyses showed that stand recruitment was positively correlated to shrub cover and soil moisture. In particular, shrub cover had a stronger effect, clearly showing that a thicker shrub layer, both bushy and/or spiny, corresponded to a higher number of yew seedlings and saplings. Secondarily, moister sites had a higher number of seedlings and saplings, showing that habitat suitability improved with higher humidity. On the contrary, recruitment was negatively correlated to browsing (both from livestock and wild animals). Our data confirm that the presence of a protective layer of shrubs is a crucial factor for seedling and sapling survival, mostly in relation to protection from summer drought and the browsing of large herbivores. Finally, guidelines for the conservation and restoration of *T. baccata* communities, referred to as the EU priority habitat 9580* (Mediterranean *Taxus baccata* woods), have been outlined.

## 1. Introduction

The Mediterranean Basin is considered one of the most altered biodiversity hotspots on Earth [1]. For a long time, many Mediterranean forest habitats suffered from human exploitation and transformations [2,3,4,5]. Consequently, only a small fraction of its primary forest vegetation, equal to 4.7%, is today preserved [6]. In wooded habitats, human disturbance caused the simplification of structure and changes in plant community composition, species abundance, and distribution [7]. It is estimated that native forests without clearly visible human activities and where ecological processes are not significantly disturbed, cover in Europe approximately 1.4 million ha, representing 0.25% of land [8].

The common yew (hereafter yew), *Taxus baccata*, has been recognized as one of the most ancient forest species in Europe, with origins in the early Miocene [9,10]. Yew grows mostly in areas where oceanic conditions exist, across Europe, the British Isles, the Mediterranean Basin, Northern Africa, the Azores and Madeira, and the Caucasus to Northern Iran [11,12,13,14,15,16]. Interestingly, in southern Italy, large Mediterranean islands, and in Hyrcanian forests (northern Iran), yew dominates or co-dominates relic stands with high biogeographical meaning [17,18]. In the Mediterranean Basin, yew is mostly a montane tree, preferentially growing in the understory of taller trees and on north-facing slopes [11].

Yew was common in many European areas during the Pleistocene interglacial periods and the early stages of the Holocene [19]. Then, the species became rarer in several parts of its distribution range, probably as a result of regression due to a change in climatic conditions during the last few millennia [19,20,21], locally disappearing or remaining isolated in small populations [22,23,24,25,26]. The main factors of reduction were recognized as human pressure, overbrowsing, poor competitive ability compared to other species, changes in water table depth, droughts, fungal infections, and dioecy-related problems [27,28,29,30,31,32,33,34]. Land use contributed to the contraction of yew habitats through logging [21,23], in combination with browsing and burning, which transformed forest landscapes and affected shade-tolerant and late-successional species such as yew [35,36]. Consequent wooded habitat fragmentation, caused by human activities, negatively impacted yew pollination by reducing its population size, since the formation of viable seeds requires the co-existence of both sexes [36]. The yew reduction has also been attributed to the overbrowsing of seedlings and saplings by herbivores, and, likewise, the scarcity of suitable sites for recruitment [21,34,37,38,39].

For the reasons illustrated above, this taxon is protected by law in several European countries [34,40,41]. Moreover, in the EU context, there are six forest habitats with yew (9120, 9180, 91J0, 9210, 9380, 9580) included in the Habitats Directive 92/43/EEC, of which two are prevalent in Italy and listed with priority status (9580*—Mediterranean *Taxus baccata* woods, and 9210*—Apennine beech forests with *Taxus* and *Ilex*) [42,43]. To conserve the natural distribution of this species, protected areas have been established and included in the Natura 2000 network as Special Areas of Conservation (SACs), which must be managed to maintain habitats in favorable conservation status [44]. To achieve this goal, the specific structure and functions necessary for the long-term persistence of habitats must be maintained and the conservation status of typical species must be favorable [35,37]. Therefore, to achieve the aim of maintaining a favorable conservation status of EU habitats, it is crucial to understand key factors affecting the “structure and function” of these habitats so that informed management decisions can be taken.

Here, we try to identify the main ecological and anthropic factors affecting yew persistence in the island of Sardinia. In the island, which is part of the southern Mediterranean range of yew distribution, the species was considered rare during the 20th and the early 21st centuries, being known in the literature from 69 sites [13,39,45,46,47]. In many sites, yew populations were represented by old-growth individuals, often very few in number, and with poor recruitment [39,48]. The species thus appeared to be under range contraction. Therefore, we focused our study on describing the structure of extant yew populations and evaluating ecological factors affecting yew regeneration, as this is a key functional process for population persistence. Regarding the latter, by means of generalized linear modeling and following the hypotheses put forward in the method Section 4.3, we evaluate the effect of wood age, sex ratio, wood closure, browsing, shrub cover, soil summer moisture, site morphology, and site slope on yew regeneration.

Overall, the main aims of this study were thus to: (i) update the present distribution of yew in Sardinia and define the main structural parameters (density, DBH, and sex ratio) of the extant populations; (ii) evaluate the most relevant factors affecting yew regeneration; (iii) based on the results, assess conservation measures useful for the future protection of this tree and its relative habitat.

## 2. Results

The data collected allowed us to report the current presence of *Taxus baccata* in 232 Sardinian sites, whereas in the other 5, the species is locally extinct. Among these sites, 168 are recorded here for the first time. The species is present in 15 different mountain areas, where it normally does not form extended communities, often being represented by single or sparse individuals. In 188 sites, the yew populations were constituted by fewer than 40 trees (110 of them with a spatial distribution of isolated individuals); another 25 sites had between 41 and 100 trees; we found more than 100 adult yews in 19 sites, 12 of which can be defined as pure or almost pure yew woodlands. In other cases, yews formed small mixed woodlands with other tree taxa (*Alnus glutinosa* (L.) Gaertn.; *Fraxinus ornus* L.; *Ilex aquifolium* L.; *Ostrya carpinifolia* Scop.; *Quercus ilex* L.; *Q.* gr. *pubescens* Willd., etc.). Among these populations, 86 had extensions smaller or equal to 0.5 ha. Another 85 sites had extensions ranging from 0.5 ha to 3 ha, whereas 52 sites were larger than 3 ha. The yew density was often low, being higher than 20 yews per hectare in 41 cases and otherwise lower. The mean DBH of Sardinian adult yews ranged from 107.8 to 734.6 mm (mean 421.2 mm), calculated on more than 80% of the observed trees. The sex ratio of more than 8300 yews showed a prevalence of males, corresponding to 55.6% of the total adult trees studied.

Among abiotic factors characterizing Sardinian yew sites, declivity averaged from 7.3 to 32.7° (mean 20°). Geologically, 77 sites were located on limestones, 76 on granites, and 61 on metamorphic rocks. In the remaining 34 sites, yews grew on different substrates (andesites, basalts, granodiorites, rhyolites, and trachyte). In total, 119 yew sites were related to wet habitats (gorges, springs, streams), whereas 118 yew sites were related to drier habitats (cliffs, rocks, slopes, and mountain tops). Regeneration was absent in 134 yew sites, although we found abundant recruitment in 9 sites.

The 40 sites selected for the GLM analysis (listed in Appendix A) showed a mean site area ranging from 0.4 to 5.1 ha (mean 2.6 ha), an average of 92 living adult yews per site (nAT in Appendix A), an average density of 10 to 64 yews per hectare (mean 37), and an average DBH ranging from 184 to 681 mm (mean 443 mm). The sex ratio was often unbalanced in favor of males (57% of the total observed samples), which were prevalent in 55.7% of sites (28 sites), but we found a slight female prevalence in 11 sites and equity of sexes in one site. Recruitment was absent in 15 sites; very rare in 3 sites; rare in 7 sites; scarce in 8 sites; frequent in 4 cases; and abundant in 3 cases.

Univariate model selection supported the effect of shrubs, sex ratio, DBH, morphology, Browsing, and Summer moisture (the first six models in Table 1) on yew regeneration, which has AICc much lower than the null model M0. The effect of shrubs (first model) is by far the strongest, with an improvement in AICc of more than 91 compared to M0 (seventh), with an estimate of overdispersion based on deviance (c_hat) = 1.27 and deviance explained = 0.66.

The full model has an estimate of overdispersion based on deviance (c_hat) = 1.19, residual deviance (39.247 with 33 df), and the sum of Pearson (36.70), which are lower than the five-percent critical value for a chi-squared distribution (47.40), thus providing an adequate description of the data.

Backward model selection provided support for model Recr ~ Shrub + Morph + Bro (first in Table 2), which is significantly different from the second-ranked (*p* LRT = 0.06134) and shows the effects of shrubs, morphology (watercourse, W), and browsing on yew regeneration. Other effects previously supported by the univariate model selection have no support with the multivariate analysis. Parameter estimates taken from the best model are Shrubs = 0.4446 (*p* = 0.0003), Morph (W) = 0.7009 (*p* = 0.0050), Bro (yes) = 0.4833 (*p* = 0.0608). The effects observed with the univariate models for Shrub (0.4028) and Morph (0.7487) are consistent with the multivariate analysis, whereas the sign of Bro is opposite in univariate (−0.7612), possibly suggesting interactions between the variables Shrubs and Bro. Additionally, the effects of SR and DBH, which seemed rather strong in the univariate analysis, are lost in the multivariate, possibly due to correlations with Shrubs (cor of Shrub vs. DBH = −0.470556; cor of Shrub vs. SR = 0.6325661).

Automated model selection performed with glmulti [49], with interactions allowed (see R script, Appendix A, for details), the same best model selected, Recr ~ Shrub + Morph + Bro, and model-averaged estimates of parameters that confirm the positive effects of Shrubs and Morph (W) provided and a negative effect of browsing on Recr (Shrubs = 0.3688, Unconditional variance = 0.0148; Morph (W) = 0.5945, UV = 0.1850, Bro (yes) = −0.0893, UV = 0.5003). Additionally, values of deviance explained showed in Table 2 evidence that Shrub has a strong effect, explaining 66% of deviance, whereas Morph provides an additional 3% and Bro 2% of explanation (see models ranked fourth, second, and first).

## 3. Discussion

### 3.1. Main Factors Affecting Yew Recruitment

Although the effects of biotic (browsing) and abiotic (soil aridity and/or moisture) factors on yew (and other trees) recruitment in Mediterranean environments are well known [21,38,50,51,52,53,54], a comprehensive analysis able to measure the relative weight of each single factor, and the interactions among factors, was still missing. Our results suggest that, albeit several ecological variables can simultaneously affect yew regeneration, shrub cover is by far the main factor positively affecting yew regeneration, especially when associated with the presence of moist sites (watercourses), whereas browsing negatively affects the establishment and survival of seedlings and saplings. The multivariate analysis carried out on multiple factors and their interactions proved to be more conservative than the univariate, selecting fewer effects. Below, we will discuss the effects supported by the multivariate analysis.

Shrub cover is by far the strongest effect. Shrubs can protect tree seedlings from browsing by providing shelter and associational resistance [55] and from drought by shading plants and facilitating mycorrhization [56,57,58,59].

Furthermore, shrub canopies create favorable conditions for yew dispersion and germination, by exerting a perch effect that attracts frugivores releasing seed rain under the canopy of fleshy-fruited shrubs, and for seedling survival by maintaining a moister environment under their canopies and avoiding or reducing browsing by large herbivores [21,53,59,60]. Yew seedlings and saplings appear to be more light-shade-tolerant than adults [60]; thus, their growth can be favored under shrubs, allowing an interaction of “shrub and yew” which facilitates juvenile yew survival. Moist and nutrient-rich sites are more suitable microhabitats for seedling survival and can be identified as favorable microsites, especially in relation to summer drought in Mediterranean environments [61], and simultaneously protect yew juveniles from large herbivores when dense (and maybe spiny) shrubs are present. The protection of yew seedlings and saplings by shrubs appears to be the most relevant factor determining the associative pattern observed in this study. Among the other positive connections, shrubs provide shelter against ungulates both for their relative unpalatability and thickness (e.g., *Arbutus unedo*, *Erica arborea*, *E. scoparia*) [55] or due to their spinescence (e.g., *Crataegus monogyna*, *Prunus spinosa*, *Rubus* gr. *ulmifolius*) [39]. As a result, in areas where a higher livestock presence was evident, juvenile yews were often absent in unprotected sites and, where some young trees were present, these assumed the typical hourglass shape, characteristic of open woodlands affected by intense browsing [62]. In these cases, shrubs protected seedlings during their first growing phases, then allowed slow but constant development and establishment of saplings. Thus, spiny and unpalatable shrubs act as natural fences to prevent herbivores from browsing seedlings, saplings, and young trees.

Soil moisture, particularly during the dry season, is another factor that our study highlighted as important for yew recruitment. It is well known that at the southern extreme of its distribution, i.e., the Mediterranean and Irano-Turanian biogeographic regions, moister and fresher sites act as niche refugia for boreal plants such as yew [11,18,63]. The presence of springs and/or watercourses nearby yew populations positively affects regeneration, very likely by providing more humid conditions for seedling establishment and survival. Yews find better environmental conditions along streams, even extending their suitable climatic and moisture requirements under the limits otherwise reached by the species in normal conditions [13,52]. On the other hand, water shortage has been considered as a limiting factor for yew recruitment [53]. In relation to the predicted consistent precipitation decrease due to climate change, water stress will be one of the challenges that yews have to face in the future, particularly in the Mediterranean Basin [53,61,63].

Browsing negatively affects yew regeneration acting mainly against seedling and sapling growth rather than seedling emergence [21,39,54]. Yew is browsed by vertebrate herbivores, although almost all parts of the plant are toxic [11,21,38,39,64,65]. In our study, among the 15 sites where a lack of recruitment was recorded, only two were totally devoid of livestock, although wildlife herbivores were locally present, meaning that apart from local climatic conditions, browsing animals appear to be a relevant factor limiting yew recruitment. The presence of browsing animals, both livestock and wildlife (cattle, goats, sheep, deer, and mouflon), was recorded in 29 sites, four of which are currently characterized by a rare to sporadic presence of yew recruitment. On the other hand, the remaining 11 sites with no evidence of browsing preserved recruitment, sometimes abundantly, confirming that a lower pressure of herbivorous mammals favors richer and more successful yew recruitment [29].

### 3.2. Implications for Conservation

Our study on the regeneration of *Taxus baccata* populations at the species’ southern range has confirmed that yew recruitment is positively correlated with woody and spiny shrubs, which protect juveniles from drought and herbivores/browsers [38,39,51], create through their shade a shield against negative climatic drivers [51], and maintain better soil conditions that facilitate the development of recruits’ root apparatus [39,52]. Therefore, actions such as maintaining or actively increasing shrub cover appear as important management activities to foster yew recruitment.

Moreover, since the yew regeneration of Mediterranean populations is higher in places with moister soil conditions and fresher microclimates, actions such as in situ seed dispersion and/or seedling plantations should be provided after prior checks of sites in order to select suitable microsites for regeneration [63].

Site management alone could not be sufficient to guarantee yew regeneration, since it was already shown that the minimum requirements for a population to be self-perpetuating are populations extended over an area of at least 0.5–3 hectares [36], with no less than 40 adult individuals and approximately equal portions of males and females [66]. Nonetheless, our study confirms that site management can be a crucial factor and should be aimed mainly at maintaining or increasing shrub cover close to sparse yew females, possibly by identifying those places where soil moisture is higher during summer droughts. These sites could be identified in places nearby springs or streams, as previously detected by Sanz et al. [52]. According to Casals et al. [54], for young yews, non-dense woodlands are most likely to represent optimal sites for regeneration and, where possible, should be managed for this purpose. Accordingly, we remark on the relationship between the density of shrub cover and yew regeneration, which consequently puts into evidence their potential facilitative interaction.

Previous studies on yew stands mostly concentrated on the negative impacts of browsing on yew regeneration (e.g., [21,39,54,64,67]). Some of these works showed that the creation of specific protected areas for the species was not sufficient, in the absence of herbivore control [11,24,64]. Sardinian yew populations are often affected by the browsing of livestock [39], to which wild animals have to be added. On sites where herbivory has long disappeared and an extended shrub layer now covers large surfaces, yew populations are increasing [68]. Similar trends were also observed in other Mediterranean areas [59]. Conversely, areas with a limited to abundant herbivore load show no or low recruitment. The circle keeps turning since a lesser presence of animals normally corresponds to higher and more uniform shrub cover and vice versa [69,70]. For this reason, among the relevant actions to perform, the reduction in and control of livestock load around yew populations should be enhanced, along with the monitoring of wildlife effects on recruitment.

Based on previous data [36,66], along with the scarce or null regeneration observed for several years during our study, some of the selected sites can be considered as non-self-perpetuating populations, posing serious questions about their active protection through in situ and ex situ actions. Specifically, it is remarkable that 38 out of 40 selected yew populations were extended over an area larger than 0.5 hectares, but only in 11 cases was this larger than 3 hectares. More alarming was the presence of 40 or more adults in only 18 out of 40 populations (Appendix A). In those cases, planting locally produced seedlings and/or saplings after the creation of shelters within selected microhabitats could be useful for the perpetuation of otherwise-threatened and prone-to-declining populations. These actions appear to be even more relevant in sites with strongly male-unbalanced populations (e.g., site 15, where only 3 females compared to 14 males were recorded).

From a strictly protection point of view, only 10 (25% of the total) of the studied sites are recorded within official maps as EU priority habitat 9580* (sensu Habitats Directive, 92/43/CEE, 35- European Commission 1992), despite the total yew sites included in the Special Areas of Conservation (SACs) being 28. Moreover, three sites are both included in SACs and are also recognized as “Regional Monuments” according to the Sardinian Regional Law 31/89; five sites are also included in the Gennargentu National Park; additionally, another two are part of the Gutturu Mannu Regional Nature Park; finally, one of these is also included in the Monte Arcosu WWF oasis.

Given the paucity of viable yew populations, it seems important to include all Sardinian sites where yews form sufficiently large stands (≥0.5 ha) within the EU priority habitat 9580* and the Natura 2000 network.

Nevertheless, the low presence of recruitment represents a future challenge for the conservation and management of threatened yew habitats, both in protected and unprotected sites. Thus, active ex situ and in situ management could prove to be fundamental for the conservation of this species at the southern extremes of its distribution range. Ex situ actions require, on the one hand, the collection of yew seeds from threatened sites, and their conservation in seed banks, from which to draw in the case of necessity. On the other hand, tree nurseries (both from seed and scion) acting as quasi-in situ sites could be created to guarantee the conservation of living material, safe from possible stresses [63].

In situ actions can be provided by planting seedlings (possibly second-years, to allow better resistance from environmental conditions) that should be protected through the creation of artificial shelters on the surrounding areas of adult yews, specifically where there is no shrub cover [34,63]. Otherwise, their plantation should be planned within sites where natural shrub shelters and soil moisture could enhance seedling survival, also in view of future possible habitat modifications connected with climate change.

## 4. Materials and Methods

### 4.1. Study System

The object of this study is yew stands growing on the island of Sardinia, which is the second largest Mediterranean island by area, with a surface of about 24,090 km², including smaller islands and islets. Sardinia is located at the center of the west-Mediterranean Basin; its latitude is N 38°51′52″ (Capo Teulada) and N 41°15′42″ (Punta Falcone) at its southern and northern extremes. The Sardinian landscape is mostly hilly, locally characterized by plateaus and plains, while the main mountains are low (average heights ca. 1000 m a.s.l.) and often isolated, rarely grouped in chains or massifs, the highest of which is the Gennargentu (1834 m a.s.l.). Sardinian substrates are heterogeneous, with the main outcrops represented by Cambrian metamorphites, carboniferous granitic batholiths, and Paleozoic limestones; in addition, local sedimentary complexes related to a Mesozoic marine transgression, Tertiary marine and volcanic depositions related to the opening of the Tyrrhenian Basin, and Quaternary alluvial deposits are present [71].

From a bioclimatic point of view, although Sardinia mostly falls within the Mediterranean Pluviseasonal Oceanic climate, some differences can be found from north to south. In particular, two macrobioclimates (Mediterranean and Temperate submediterranean), four classes of continentality (from weak semi-hyperoceanic to weak subcontinental), eight thermotypic horizons (from lower thermomediterranean to upper supratemperate), and seven ombrothermic horizons (from lower dry to lower hyperhumid) define the entire region [72,73].

The Sardinian forest extension was considerably reduced starting from Punic and Roman times as a result of massive deforestation for timber and extensive agricultural and pastoral activities, combined with fire to maintain pastures [74,75,76,77]. In several mountain areas that still remain relatively isolated and scarcely populated, the island preserves the wilderness of natural environments with difficult access, which are therefore relatively well preserved [78]. Relic yew stands can be found in some of these less disturbed areas, on gorges, sinkholes, and mountain slopes and tops, and are very interesting examples of woodlands related to specific edaphoclimatic conditions [13,45,46,47,48,79]. From a floristic–vegetational point of view, Sardinian woodland communities dominated or co-dominated by *Taxus baccata* are referred to two main associations (*Cyclamino repandi-Taxetum baccatae*, *Polysticho setiferi-Taxetum baccatae*) belonging to the Cyrno-Sardinian endemic alliance *Lathyro veneti-Taxion baccatae* [80] Moreover, two subassociations of oak woodlands (*Glechomo sardoae-Quercetum congestae taxetosum baccatae* and *Saniculo europaeae-Quercetum ilicis taxetosum baccatae*) are known [13].

### 4.2. Data Collection

We recorded all localities where *Taxus baccata* individuals were reported in the literature (e.g., [13,45,46,47,81]). In addition, we found further information about yew distribution in Sardinia through interviews with local people (forestry workers, shepherds, environmental guides), by consulting maps issued by the Italian Military Geographic Institute (IGMI, scale 1:25,000), and locating toponyms recalling the yew, i.e., Tassu/Tassos/Tassi; Eni/Enis; Longufresu/Longuvresu; Niberu/Nibberu—this latter meaning mostly juniper, but locally it is used for yews. Then, we created a dataset containing all yew localities found from different sources. Finally, we carried out field surveys (from 2015 to 2021) to collect ecological and structural data and to evaluate the current status of known yew populations. All confirmed localities were geo-referenced and recorded into a GIS by using the Open-Source Geographic Information System Quantum GIS (QGIS 3.18). Then, we generated a distribution map with all the yew sites found, updating the current Sardinian distribution.

Later, we selected 40 sites from the entire distributional range of the species on the island (Figure 1; Appendix A). The sites were selected based on geographical, ecological, and structural features. From a geographical point of view, we selected a proportional number of sites from each mountain area, thus representing all mountain sectors. The average elevation of selected stands was 620–1250 m a.s.l. Among the studied stands, 14 can be classified within the phytosociological association *Cyclamino repandi-Taxetum baccatae*; four within the *Polysticho setiferi-Taxetum baccatae*; eight within the subassociation *Saniculo europaeae-Quercetum ilicis taxetosum*; and one within the *Glechomo sardoae-Quercetum congestae taxetosum*. The other 13 stands were not assigned to any phytosociological unit. Almost all selected stands grew in the Mediterranean Pluviseasonal Oceanic bioclimate, whereas four stands grew in the Oceanic temperate one. These stands were characterized by being part of woodlands where yews had a relevant presence, with at least 15 adult trees per site (only site 32—which is the southernmost—excepted, with 11 living samples).

Yew regeneration (abundance of seedlings and abundance of saplings) was estimated in two to five 20 × 20 m plots at each site (Appendix A) unless the site area was smaller as to allow a total sampling. Plots were sampled both within the stands and in clearings/edges of woodlands to infer potential diversity between places with different light availability. The abundance of juveniles was subdivided into six classes: absent (0); very rare (1 = <1/ha); rare (2 = 1–10/ha); scarce (3 = 11–50/ha); frequent (4 = 51–100/ha); and abundant (5 = >100/ha).

### 4.3. Factors Affecting Yew Recruitment

At each site, we measured the following ecological and structural variables (Appendix A), to which yew regeneration was hypothesized to be related:(1)Wood age; according to previous studies, yews start reproducing at ages comprised between 35 and 70 years [11]; consequently, the youngest formations are not capable of an efficient system of sexual reproduction. On the contrary, older trees can produce abundant quantities of pollen and seeds, allowing easier dispersal through frugivorous animals. Wood age was approximated by average diameter at breast height (aDBH), measured on as many as possible (often all) individuals counted per site. We selected sites with average DBH ranging from 13 to 130 cm, to infer possible differences between young formations and older stands.(2)Sex ratio [SR = Females/(Females + Males)]; the reproductive ecology of dioecious species is important in the understanding of different dynamics related to the spread of such species [17,66]. The importance of sex ratio in the analysis of population evolution is related to the fact that in many dioecious species, male prevalence was highlighted, especially under stressful conditions [82]. Females are subject to higher stresses, due to the major effort put on the reproductive phases, resulting in a diminished structural increase and higher mortality when under stress [83,84,85]. Plant populations characterized by wind pollination and abiotic dispersal were found to have more often male-biased sex ratios [86]. Recent studies showed how the growth rate of *T. baccata* females was lower than that of males, together with a higher water request for females [66,85,87]. A Mediterranean climate regime would favor male prevalence more than other European regions [17]. Moreover, it was supposed that older populations were male-sex-biased [17,60,66]. Consequently, we supposed that sex-biased populations were less capable of reproduction. The sex ratio was detected by analyzing all possible individuals in the field, through the observation of pollen strobiles on male trees and arils on females. Although pollen and arils occur in different periods of the year, the contemporary observation of sexual elements in the same populations was possible by searching their remnants both on the branches and on the ground, under the trees’ canopies. In the case of larger populations, more checks were carried out.(3)Wood closure (Clos); shading could affect seed germination and seedling establishment [17,32]. The structure of older yew stands could affect yew regeneration by impeding or reducing the growth of seedlings, mostly where canopies of conspecific trees do not allow a viable light quantity [17,22,32,88]. The closure was estimated as % ground cover of canopy projections.(4)Browsing (Bro = yes or no); the action of herbivores is well known to negatively affect the survival and establishment of seedlings, and thus the regeneration of yews [39,64]. Browsing was assessed by recording any evidence related to current impacts attributable to livestock and wild animals and/or their signs (e.g., physical presence observed in the field, excrements, tracks, fur/wool on the bark and on branches, decortication by deers’ antlers, etc.).(5)Shrub cover (Shrub + Spi); shrubs could protect seedlings from drought and browsing, thus promoting the regeneration of trees [38,39]. Here, total shrub cover was measured as the sum of abundance indexes of spiny shrubs (Spi, ranging from 0 to 5) and other shrubs (Shr, 0–5), meaning 0 = absent; 1 = very rare; 2 = rare; 3 = sporadic; 4 = frequent; and 5 = abundant.(6)Soil summer moisture (Ssmo = Yes or No); Mediterranean habitats are characterized by more or less long dry periods that could negatively affect yew recruitment [11]. Water availability has been considered a limiting factor for yew regeneration in the southern range of the species [53]. Moreover, it has been demonstrated that drought stress reduces leaf size and chlorophyll content, whereas it increases secondary metabolites and antioxidants, and their negative effect on plant morphology (stem length, leaf size, vegetative growth) and physiology (Pn, gs) [89]. Ssmo was assessed by observing in the field the soil moisture of the sites at the peak of the dry season.(7)Site morphology (Morph = Slope or Watercourses); rockiness and exposure to climatic factors (sun irradiation, winds, precipitations, frost) could affect microhabitat features, particularly in relation to microclimate (dryness and moisture). In particular, watercourses have higher moisture, and it is supposed that they allow a better regeneration than slopes [11]. Half of the selected sites (20) were chosen at dry conditions (slopes) and 20 further sites at moist localities (streams, gorges, springs).(8)Site slope (Pend); declivity was supposed to be important for the persistence of more natural conditions of the wooded formations [79] and for longer-lasting moisture [52]. Slopes were also found to be potentially relevant for seed dispersal in combination with water in some yew populations [90]. Site slope was quantified as average declivity and was calculated as the average of different slopes detected using an inclinometer.

### 4.4. Statistical Analysis

The state variable of interest, Recruitment (Recr), was expressed as a count (range = 0–9). The analysis was thus performed using generalized linear models (GLMs) with Poisson error structure, with software R version 4.1.1 [91]. Considering the limited sample size (n = 40), a univariate analysis was first performed to screen for supported effects. That is, each effect was evaluated at a time, according to the following model (see R Script, Online Resource 1, for details): MEffect1 = glm (formula = Recr ~ Effect1, family = Poisson).

Univariate effect models were compared with the null model, M0 = glm (formula = Recr ~ 1, family = Poisson), by means of AICc [92] and a likelihood ratio test (LRT) [93].

Models that, compared to the null, had ΔAICc higher than two, or that differed significantly from M0 according to the LRT, were considered for multivariate analysis. Effects supported by the univariate model selection were then combined in a general multivariate model and evaluated with backward model selection, as below:MFull < -glm(formula = Recr ~ Shrub + SR + DBH + Morph + Bro + Ssmo, family = Poisson)

General model simplification proceeded by removing effects that were non-significant in the multivariate analysis, starting from higher *p* values. Multivariate model selection was also performed using the automated model selection routine of R package glmulti [49] (see R script, Online Resource 1, for details).

Model selection tables based on AICc were developed using the R package AICcmodavg [94], whereas LRTs were performed using the R package lmtest [95]. By means of multivariate model selection, simultaneous effects of variables selected with the univariate screening could be evaluated, considering also possible interactions.

The explanatory power of models was estimated as Deviance explained:DE = (deviance(M0)-deviance(MEffect))/deviance(M0)).

## 5. Conclusions

After a seven-year-long research study across Sardinia, the current distribution of common yew (*Taxus baccata*) on the island has been updated. Historical data and literature reported 69 sites where yew was previously known, while today they are 237, 5 of which were previously known and are today locally extinct. Among the extant yew growing sites, we selected 40 populations in order to represent the entire island. Then, we defined the structural parameters of the analyzed yew populations, such as density, DBH, sex ratio, and the presence/absence of recruitment. The focus of our work was the evaluation of the most relevant factors affecting yew recruitment. By means of generalized linear models, we analyzed biotic and abiotic factors (aspect, browsing, morphology, shrub cover, slope, soil moisture) which could affect yew regeneration. We developed a comprehensive analysis to measure the weight of different ecological variables and the interactions among them. The results of this work highlighted the strong positive interaction between shrub cover, both bushy and spiny, and yew recruitment through a multivariate analysis. We also evidenced the positive effects of soil moisture on yew recruitment, while negative effects were correlated to browsing (both from livestock and wild animals). We discussed how shrub cover, especially if combined with moister soils, can act as a protective layer, which enhances the capability of seedlings and saplings to survive summer drought and the browsing of large herbivores. Finally, our study suggests that to achieve yew conservation in its southern range, it is important to: (1) maintain or actively increase shrub cover; (2) select moist microsites for yew restoration; (3) reduce or control herbivores; (4) collect and conserve yew seeds in seed banks; (5) plant and shelter seedlings; and (6) include extant yew populations in the Natura 2000 network.

## Figures and Tables

**Figure 1 plants-12-01819-f001:**
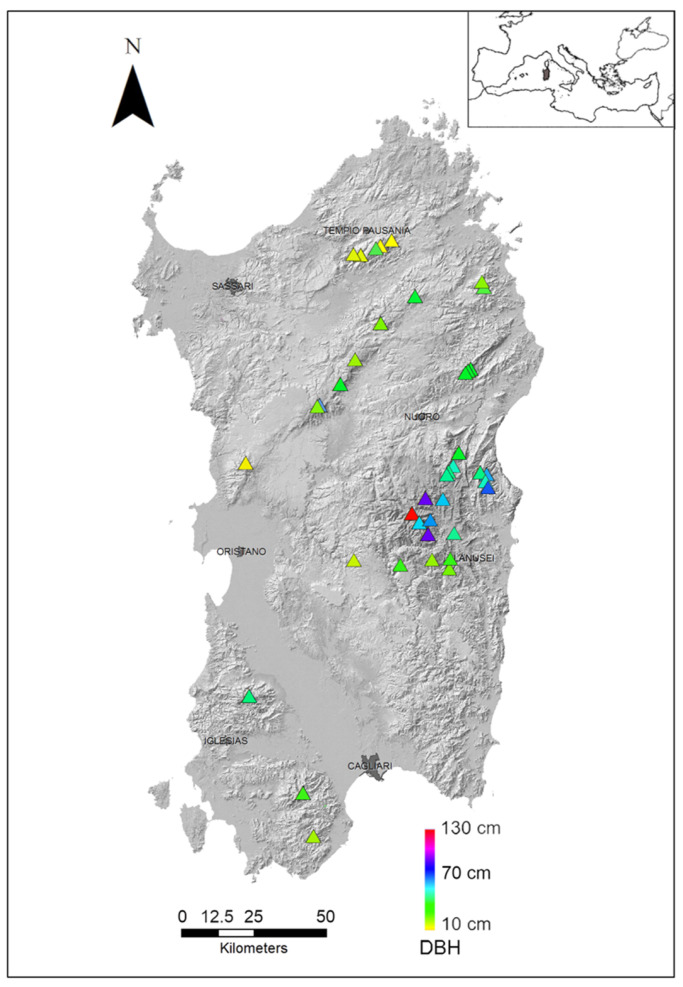
Distribution map of the 40 Sardinian yew stands selected in this study and their DBH average ranges. Painted triangles show the different DBH from a minimum of about 10 cm (yellow), representing younger formations, to a maximum of 130 cm (red), which represent the older ones. Green and bluish triangles indicate the intermediate DBH ranges (i.e., adult and mature formations).

**Table 1 plants-12-01819-t001:** Variables affecting common yew regeneration supported by the univariate analysis. Recr = recruitment; Shrub = shrubs; SR = sex ratio; DBH = Diameter at Breast Height; Morph = morphology; Bro = browsing; Ssmo= soil summer moisture; Clos = closure of canopies; Pend = declivity. Acronyms: K = number of model parameters; AICc = small sample correction of Akaike’s information criterion; ∆AICc= model AICc—minAICc; LL = model log-likelihood; Dev. Expl. = deviance explained.

Rank	Model Structure	K	AICc	∆AICc	LL	Dev. Expl.
**1**	Recr ~ Shrub	2	130.2244	0.0000	−62.9501	0.6606
**2**	Recr ~ SR	2	183.2421	53.0176	−89.4589	0.2877
**3**	Recr ~ DBH	2	187.7552	57.5308	−91.7154	0.2560
**4**	Recr ~ Morph	2	209.8880	79.6635	−102.7818	0.1003
**5**	Recr ~ Bro	2	211.1093	80.8849	−103.3925	0.0917
**6**	Recr ~ Ssmo	2	213.5470	83.3225	−104.6113	0.0746
**7**	M0: Recr ~ 1	1	221.9372	91.7127	−109.9159	0.0000
**8**	Recr ~ Clos	2	223.8796	93.6551	−109.7776	0.0000
**9**	Recr ~ Pend	2	224.1561	93.9317	−109.9159	0.0000

**Table 2 plants-12-01819-t002:** Variables affecting common yew regeneration supported by the multivariate analysis. Shrub = shrubs; Morph = morphology; Bro = browsing; DBH = diameter at breast height; Ssmo= soil summer moisture; SR = sex ratio. Acronyms: K = number of model parameters; AICc = small sample correction of Akaike’s information criterion; ∆AICc = model AICc—minAICc; wi = Akaike’s weight; LL = model log-likelihood; Dev. Expl. = deviance explained.

Rank	Effects on Yew Regeneration	K	AICc	∆AICc	wi	LL	Dev. Expl.
**1**	Shrub + Morph + Bro	4	127.1372	0.0000	0.4248	−58.9972	0.7162
**2**	Shrub + Morph	3	128.1617	1.0245	0.2545	−60.7475	0.6916
**3**	Shrub + DBH + Morph + Bro	5	129.3459	2.2087	0.1408	−58.7906	0.7191
**4**	Shrub	2	130.2244	3.0872	0.0907	−62.9501	0.6606
**5**	Shrub + DBH + Morph + Bro + Ssmo	6	131.4385	4.3013	0.0494	−58.4465	0.7240
**6**	Shrub + Bro	3	132.5451	5.4079	0.0284	−62.9392	0.6608
**7**	Shrub + SR + DBH + Morph + Bro + Ssmo	7	134.3926	7.2554	0.0113	−58.4463	0.7240
**8**	Morph + Bro	3	208.3445	81.2073	0.0000	−100.8389	0.1277
**9**	Morph	2	209.8880	82.7508	0.0000	−102.7818	0.1003
**10**	Bro	2	211.1093	83.9721	0.0000	−103.3925	0.0917
**11**	None	1	221.9372	94.8000	0.0000	−109.9159	0.0000

## Data Availability

The datasets generated during and/or analyzed during the current study are available from the corresponding author upon reasonable request.

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
