# Peer review of "Shrub Cover and Soil Moisture Affect Taxus baccata L. Regeneration at Its Southern Range"

_plants, 2023, doi:10.3390/plants12091819_

Round 1
Reviewer 1 Report
General comments
I have read the manuscript: Entitle: Shrub cover and soil moisture affect Taxus baccata L. regeneration at its southern range written by Giacomo Calvia et. al., for publication of Plants MDPI. In this study, the author selected 40 sites, located in 14 different mountain chains, characterized the adult yews with average Diameter at Breast Height (DBH) from 13 to 130 cm. Author found the positively correlated to shrub cover and soil moisture and was negatively correlated to browsing (both from livestock and wild animals).
The overall research is well conducted but author should significantly improve this manuscript for journal acceptance. Finally, guidelines for conservation and restoration of T. baccata communities, referred to the EU priority habitat 9580 have been outlined. In this sense this manuscript is much valuable. However, I found a lack of story connection and some lack of potential references (some I suggested below). Overall after I evaluate and request the author for this manuscript as a “MAJOR REVISION”. If author well address the comments and improve the manuscript this article may accept for the publication.
Major Suggestions
1) Abstract: Abstract should be more informative and should be present the new insight of the research clearly and concisely. The result section should be more robust, the present results section in the abstract is comparatively shallow. Please consider that the short and concise, more informative.
2) Hypothesis of the study: The author well presented the research aim or objective of the study clearly in Ln. 76-80, but research hypothesis is not clearly mentioned and is not connected with the objectives. The research hypothesis should be very clear and connected each other (with objectives) because without appropriate literature, questions, or hypotheses the entire introduction section will not be clear.
3). Introduction: Author started the background of the Mediterranean basin and forest habitat, climate change issue and its ultimate impact on the vegetation, which is much appreciated. However, author should refer and mainly emphasis some literatures related to the drought/temperature stress related articles as a reference in introduction section. Author should mention that “Drought stress reduces the leaf size, reduction of Chl. increases the secondary metabolites and antioxidants, their negative effect on morphology (stem length, leaf size, vegetative growth) and physiology (Pn, gs)”. This article https://doi.org/10.1016/j.foreco.2020.118099 well describe the negative effect of drought stress.
Some other comments
4) Line 149: Please remember that in scientific writing each figure is independent, and it should speck all the component of the figure clearly.
5) Line 160: I am also quite not satisfied about the way to present the discussion section. Please make the separate sub-title and describe them separately with the appropriate references.
6) Line 160: Author should present the very closely information and discuss them only by referring the potential references while deal with the vegetation coverage, morphological sections and allocation of biomass and identify the diameter and breast height. https://doi.org/10.1016/j.scitotenv.2021.146466
7) Line 293 after (Conclusion): Author should be mentioned the independent conclusion section instead of the summary. Further should not be repetitive in the abstract or a summary of the results section. I would love to read striking points and take-home messages that will linger in the readers’ minds. What is the novelty, how does the study elucidate some questions in this field, and the contributions the paper may offer to the scientific community?
8) Line 464 (References): please double-check the citations, their style, spell check, and other grammatical errors. moreover, the author should cut the old and less matching literature and include the latest literature some of them are above.
Good Luck!
Author Response
Referee 1:
The overall research is well conducted but author should significantly improve this manuscript for journal acceptance. Finally, guidelines for conservation and restoration of T. baccata communities, referred to the EU priority habitat 9580 have been outlined. In this sense this manuscript is much valuable. However, I found a lack of story connection and some lack of potential references (some I suggested below). Overall after I evaluate and request the author for this manuscript as a “MAJOR REVISION”. If author well address the comments and improve the manuscript this article may accept for the publication.
Major Suggestions
Abstract: Abstract should be more informative and should be present the new insight of the research clearly and concisely. The result section should be more robust, the present results section in the abstract is comparatively shallow. Please consider that the short and concise, more informative.
Answer: done.
Hypothesis of the study: The author well presented the research aim or objective of the study clearly in Ln. 76-80, but research hypothesis is not clearly mentioned and is not connected with the objectives. The research hypothesis should be very clear and connected each other (with objectives) because without appropriate literature, questions, or hypotheses the entire introduction section will not be clear.
Answer: done.
Introduction: Author started the background of the Mediterranean basin and forest habitat, climate change issue and its ultimate impact on the vegetation, which is much appreciated. However, author should refer and mainly emphasis some literatures related to the drought/temperature stress related articles as a reference in introduction section. Author should mention that “Drought stress reduces the leaf size, reduction of Chl. increases the secondary metabolites and antioxidants, their negative effect on morphology (stem length, leaf size, vegetative growth) and physiology (Pn, gs)”. This article https://doi.org/10.1016/j.foreco.2020.118099 well describe the negative effect of drought stress.
Answer: done. Since part of the requested text changes were already present in the Materials and Method section, we added a reminder to it and created a new sub-chapter: 4.3 Factors affecting yew recruitment. We also added the article suggested by the referee, but for the reason explained above it fitted better in the M&M section.
Some other comments
Line 149: Please remember that in scientific writing each figure is independent, and it should speck all the component of the figure clearly.
Answer: done.
Line 160: I am also quite not satisfied about the way to present the discussion section. Please make the separate sub-title and describe them separately with the appropriate references.
Answer: done.
Line 160: Author should present the very closely information and discuss them only by referring the potential references while deal with the vegetation coverage, morphological sections and allocation of biomass and identify the diameter and breast height. https://doi.org/10.1016/j.scitotenv.2021.146466
Answer: the addition of this part was not possible since our discussion is structured for explaining the results and there was no other result than ecological. We did not study physiology of plants and the addition of such part should represent a forcing.
Line 293 after (Conclusion):Author should be mentioned the independent conclusion section instead of the summary. Further should not be repetitive in the abstract or a summary of the results section. I would love to read striking points and take-home messages that will linger in the readers’ minds. What is the novelty, how does the study elucidate some questions in this field, and the contributions the paper may offer to the scientific community?
Answer: done
Line 464 (References): please double-check the citations, their style, spell check, and other grammatical errors. moreover, the author should cut the old and less matching literature and include the latest literature some of them are above.
Answer: we have revised the references, but no cuts were made, given that all the cited literature was useful and matched with our work. Nonetheless, we have added the references suggested by the reviewer.
Reviewer 2 Report
I read with great interest the article titled: "Shrub cover and soil moisture affect Taxus baccata L. regeneration at its southern range”, submitted to the special issue of Plants (MDPI).
English yew (Taxus baccata L.) is a rare and endangered tree species in many European countries. The restricted occurrence is mainly due to intensive human land use and the effects of forest management which changed the structure and species composition of the remaining forests. The main reasons for the decline of yew are widespread deforestation, light competition, selective felling of yew, and browsing by herbivores. Research studies on endangered tree species usually focus on the environmental conditions of the population, the regeneration status, and the causes of declination.
In this study, the authors update the present distribution of Taxus baccata in Sardinia (Italy) and define the main structural parameters (density, DBH, and sex ratio) of the extant populations, next evaluate the most relevant factors affecting yew regeneration and assess conservation measures useful for the future protection of this tree and the habitat it identifies. Research results obtained by the authors confirm that a protective layer of bushy and spiny shrubs is a crucial factor for seedling and sapling survival, mainly in relation to protection from summer drought and the browsing of large herbivores.
The work submitted for review generally is well-written, and the methodology and conclusions are scientifically sound. I found the paper interesting. The investigations are extensive. The presentation of results is clear and attractive. As such, the manuscript is potentially interesting to a relatively broad readership covering fields like evolutionary and conservation biology. My comments mainly relate to relatively minor issues of interpretation and writing. These comments do not influence a positive impression of the article.
Some minor notes:
(1) Please add a legend to Figure 1. Distribution map of the Sardinian selected yew stands and their DBH average ranges. The lack of explanations of the graphic symbols makes it difficult to analyze.
(2) Please add a conclusion chapter summarizing the most important results of the research conducted.
In my opinion, this is an interesting article, I recommend publishing it with major corrections.
Author Response
Response to Reviewer 2 Comments
Point 1: Please add a legend to Figure 1. Distribution map of the Sardinian selected yew stands and their DBH average ranges. The lack of explanations of the graphic symbols makes it difficult to analyze.
Response 1: done. I hope now it is better.
Point 2: Please add a conclusion chapter summarizing the most important results of the research conducted.
Response 2: done.

Reviewer 3 Report
Dear Authors,
Analyzing the manuscript, I see that it is well written and has a good structure. The number of errors is few, and the language is understandable. The problem statement and the aims are clear. Based on these, I recommend the manuscript for publication after correcting the minimal marked errors.
The subfigure (Europe) is too small, as are the legends in the Fig. 1.
The terms 'in situ' and 'ex situ' should be written in italics. Please make corrections throughout the manuscript.
Author Response
Response to Reviewer 3 Comments
Point 1: The subfigure (Europe) is too small, as are the legends in the Fig. 1.
Response 1: done. I hope now it is better.
Point 2: The terms 'in situ' and 'ex situ' should be written in italics. Please make corrections throughout the manuscript.
Response 2: done.

Round 2
Reviewer 1 Report
Dear Author
I have read the revised manuscript plants-2349330. Entitled: Shrub cover and soil moisture affect Taxus baccata L. regeneration at its southern range for publication in Plants. This is the second submission made by the author. The author addressed all the questions and suggestions that I raised the issue in the review of the original manuscript. I satisfy the author’s revisions. Author improves their hypothesis and well connected with the research objectives in this time. This manuscript improved the flow of writing, which was comparatively shallow in the original version but in this revised copy author very well addressed all the quarries and suggestions. Before accepting this manuscript if there is anything needed to be revised by the author, especially English grammar, or spell check, I request this manuscript is currently in “Minor Revision” and the author may correct any further grammatical errors (if any) the author may improve in this stage.
Thank you.
Reviewer 2 Report
The authors have adequately addressed my comments and suggestions for improvement and/or correction. I have no further comments. I recommend publishing this article.
The text is understandably written, but it can be improved.